# Mind the Wound!—Fruit Injury Ranks Higher than, and Interacts with, Heterospecific Cues for *Drosophila suzukii* Oviposition

**DOI:** 10.3390/insects12050424

**Published:** 2021-05-09

**Authors:** Renate Kienzle, Marko Rohlfs

**Affiliations:** Population and Evolutionary Ecology Group, Institute of Ecology, University of Bremen, 28359 Bremen, Germany

**Keywords:** host selection, host cues, preference hierarchy, avoidance

## Abstract

**Simple Summary:**

By using information from the environment, insects make decisions about where to lay eggs. If we know the environmental cues used by insects, we can help in controlling insect pests. The fruit fly *Drosophila suzukii* is one such pest that attacks numerous fruit varieties and causes great economic damage worldwide. In this study, we investigated which fruit characteristics the flies prefer for egg laying. In laboratory experiments we gave *D. suzukii* females the choice between different types of blueberries: (1) intact; (2) artificially wounded; (3) wounded and infested with eggs of other fruit-fly species; and (4) intact but exposed to another fruit fly species that had not laid eggs but had defecated on the fruits. The *D. suzukii* females preferred the different fruit types for egg laying in the following order: (1) > (4) > (2) > (3). The cues of other fly species (4) have already been suspected to deter egg laying in *D. suzukii*. Here we identified a new cue—the “wounding factor” of injured fruits (2)—that deterred egg laying even more than the cues of other fly species. Knowing the chemical or physical properties of this factor could help manipulate the behavior of the flies to protect fruit from this pest.

**Abstract:**

*Drosophila suzukii* is a globally distributed insect that infests many economically important fruit varieties by ovipositing into ripening fruits. The mechanisms underlying host selection, in particular the fly’s preference for fresh, intact, and competitor-free fruits, are only partially understood. We hypothesize that *D. suzukii* females use cues of different fruit properties to rank potential host fruits in a hierarchical manner. We created four naturally occurring fruit (blueberries) categories: (1) intact; (2) artificially wounded; (3) wounded + containing eggs of different *Drosophila* species; and (4) intact + exposed to *D. melanogaster*. Individual *D. suzukii* females were offered several fruits in different two-way combinations of the fruit categories. Females showed a robust oviposition preference for intact vs. wounded + infested fruits, which was even stronger compared to the intact–wounded combination. Females preferred ovipositing into intact vs. intact + exposed blueberries; however, they preferred intact + exposed over wounded blueberries. This implies a hierarchical host preference in *D. suzukii*, which is determined by heterospecific cues (possibly fecal matter components) and an unknown “wounding factor” of fruits.

## 1. Introduction

Throughout the animal kingdom, cues of potential predators and competitors play a critical role in habitat selection [1,2,3,4]. Avoidance of heterospecific cues by the fruit pest *Drosophila suzukii* is currently discussed as a possible mechanism that reduces interspecific competition with faster developing *Drosophila* spp. [5,6]. Unlike related *Drosophila* fruit flies (e.g., *D. melanogaster*) *D. suzukii* possesses a strongly sclerotized ovipositor, with which females can pierce the skin of, and lay eggs into, numerous wild and domesticated fruits [7]. The ability to infest still ripening and undamaged fruits, e.g., cherries, blueberries, strawberries, etc., makes *D. suzukii* a serious insect pest [8] that is expanding its range worldwide [9]. The reasons for its massive spread are poorly understood, yet climatic change, global goods transportation, and the ability to use an extremely wide range of domesticated and wild fruits as egg-laying sites are likely to play an important role. On a local scale, egg-laying avoidance in response to specific, e.g., heterospecific, cues might facilitate *D. suzukii* dispersal within and among fruiting bushes or trees. As the decision to lay eggs is influenced by numerous cues that act in combination rather than individually, we require a better understanding of the interplay and hierarchy of the different cues [10] that drive the spatial distribution of *D. suzukii* through egg-laying decisions.

Shaw et al. (2018) [5] observed that groups of *D. suzukii* females tend to avoid egg laying on an artificial fly culture medium infested with eggs and larvae of heterospecific *Drosophila* (*D. melanogaster*). In part, similar observations were made by Kidera et al. (2020) [6], who let different *Drosophila* species oviposit on an artificial fruit medium (grape juice agar) and subsequently allowed groups of *D. suzukii* females to make their oviposition choice. Here we argue that the use of an artificial substrate and fly groups in confined spaces can be a serious problem for identifying a hierarchical order of oviposition preferences among different potential host fruit categories.

Firstly, using an artificial substrate in combination with heterospecific cues ignores the widely accepted fact that *D. suzukii* prefers fruits with intact fruit skin, e.g., [11]. An artificial agar-based substrate makes it impossible to manipulate a factor that almost always comes along with the presence of potential heterospecific competitors in the field, namely the wounding of fruits. Most other *Drosophila* species that could become larval competitors in decaying fruits critically depend on cracks or wounds extending over a large area in the skin of fruits to insert their eggs into the exposed fruit flesh. We consider cues associated with injured fruits to be particularly relevant, as we recently found that wounded blueberries were the least preferred fruit category in *D. suzukii* oviposition [11].

Secondly, most laboratory-based choice experiments use groups of *D. suzukii* females, although there is no evidence that the animals make their oviposition decisions in aggregations. Using groups of flies may enforce competition among females, which can influence individual decision making within such a group [12]. For example, competing conspecifics may induce a state of resource, and hence time limitation, in individual females that may cause them to lay their eggs in fruit categories they would avoid if they were not forced to do so. This means that the use of groups of flies in experiments may mask the decision making of individual flies and thus not necessarily reflect the situation in the field, where *D. suzukii* has not been found to aggregate [13].

To overcome these potential pitfalls in understanding the host selection of *D. suzukii* we quantified the results of egg-laying decisions made by individual *D. suzukii* females [11] in response to heterospecific and fruit-wounding cues. We used a field-related host fruit system (blueberries) and investigated the effects of heterospecific cues and variation in the availability of different host fruit categories, i.e., intact vs. wounded fruits. We found that wounding cues rank higher than heterospecific cues for *D. suzukii* egg distribution decision, but it is the combination of these cues that is most effective in deterring females from laying eggs. The new “wounding factor” in the cue hierarchy for *D. suzukii* oviposition could become relevant for controlling the flies’ dispersal and population growth.

## 2. Materials and Methods

### 2.1. Fly Species and Culture Conditions

*Drosophila melanogaster*, *Drosophila simulans*, and *Drosophila subobscura* were used to test whether possible cues indicating the presence of potential interspecific competitors influence the oviposition decisions of *D. suzukii*. The *D. suzukii* population used in this study originated from flies that emerged from elderberries collected in September 2016 near Kiel (Northern Germany). The *D. melanogaster* population emerged from fruits collected in 2003 in Kiel (Northern Germany). The *D. subobscura* population were established from flies collected in Göttingen (Germany) in 2016. The *D. simulans* were purchased from the National *Drosophila* Stock Center at Cornell University, NY, USA, but they are also commonly present in Europe. The selected species oviposit into smashed fruits and are widespread in the area from which the *D. suzukii* population originated. These three common species were chosen to test for species-specific effects on the egg-laying decision of *D. suzukii*.

To enhance the motivation to accept blueberries as an oviposition site, all species were reared on blueberries prior to experimentation. For the rearing, frozen blueberries were thawed and crushed to enable egg laying for all *Drosophila* species, except *D. suzukii*. Egg-infested berries were transferred to a standard *Drosophila* culture medium [14] for the larvae to develop in a low-competition environment. *D. suzukii*, *D. melanogaster*, and *D. subobscura* were reared at 20 °C ± 1 and *D. simulans* at 25 °C *±* 1, at a 16 h light cycle for all species. All species were kept at a population size of approximately 200 flies (10–15 days old) in custom-made population cages (22 liters). The flies had ad libitum access to water and decaying medium. In addition, *D. melanogaster*, *D. simulans*, and *D. subobscura* were fed with a mixture of sugar and dried baker’s yeast (1:1) to ensure survival and egg maturation. *D. suzukii* were offered fresh and intact blueberries every two days to ensure oviposition experience.

### 2.2. Host Fruit Categories and General Experimental Design

We used fresh, organically grown blueberries purchased from a local supermarket in Bremen, Germany, to prepare following host fruit categories for *D. suzukii*: (1) intact; (2) intact but artificially wounded; (3) wounded and infested with eggs of other fruit-fly species; and (4) intact but exposed to another fruit-fly species. For the fresh and “intact” category the fruits remained untreated, while “wounded” fruits were created by removing the calyx of the blueberries with a scalpel [11]. Two methods were used to generate fruits that contained cues of potential competitors: (1) wounded fruits were offered to different heterospecific *Drosophila* in their population cages (Experiment 1), where the flies laid eggs in the wounds and (2) intact fruits were exposed to a population of *D. melanogaster* (Experiment 2), where the flies defecated on the fruits but did not lay eggs. The general experimental setup was the same as described in [11]. Twelve fruits were arranged in a grid of 4 × 3, with each fruit being circa 1.5 cm apart, in 1 L plastic boxes (Pro-Pac Ostendorf Plastic, Vechta, Germany). Only one five- to ten-day-old *D. suzukii* female was released into each of these arenas, each of which was a replicate. At 4 pm ± 1 h (beginning of the flies’ egg-laying period) the arenas were prepared and subsequently incubated for 24 ± 1 h in a climate chamber at 25 °C ± 0.5 and a 16 h light cycle. On the next day, the females were removed and the eggs per fruit were counted for each arena individually. Replicates with only 0 to 3 eggs were excluded from the statistical analysis.

### 2.3. Experiment 1: Drosophila suzukii Egg-Laying Response to Fruits with Heterospecific Egg-Laying Cues

We tested whether cues related to egg infestation by heterospecific and potentially competing *Drosophila* turn blueberries into a host that ranks lower than intact or wounded fruits. Wounded blueberries that had 5 to 30 freshly laid eggs were used in the experiments. To avoid larvae hatching having unwanted effects on the egg-laying decision of *D. suzukii*, we carefully removed the eggs with dissecting needles. To test whether the proposed response of *D. suzukii* to the heterospecific cues changes due to limitations in the availability of the preferred fresh and intact fruits [11], we tested different relative abundances of intact fruits ranging from 50% (6 intact/6 treated) to 33% (4 intact/8 treated) and to 17% (2 intact/10 treated). In summary, intact blueberries were combined with four fruit treatments (wounded only or infested by *D. melanogaster*, *D. simulans*, or *D. subobscura*) at three different relative abundance levels. Each treatment–abundance combination was replicated 26 times. Replicates with only 0 to 3 eggs laid by *D. suzukii* were excluded from the statistical analysis.

### 2.4. Experiment 2: Drosophila suzukii Egg-Laying Response to Heterospecific and Fruit-Wounding Cues

In Experiment 1, we could not find any effect of the number of heterospecific eggs previously present in the treated fruits to explain how *D. suzukii* distributes its eggs (see the Results section). Thus, we assumed that the cue *D. suzukii* responds to acts independently of egg laying, e.g., the cue is contained in the fecal matter deposited during egg laying. This finding enabled us to test whether heterospecific and wounding cues act independently and rank differently in the *D. suzukii* egg-laying response. To test this, fresh and intact blueberries were exposed to a *D. melanogaster* population for approximately 1.5 h. During this period, we observed numerous flies entering the fruits and depositing feces onto the skin of the fruits without egg laying. Six of these “exposed” fruits were randomly combined with either six intact or six wounded blueberries. In a third treatment, the same numbers of intact and wounded fruits were combined. In a fourth treatment, six wounded only berries were combined with six wounded berries plus heterospecific cues; see Experiment 1 for the latter fruit treatment. A total of 35 replicates per treatment combination were set up. Replicates with only 0 to 3 eggs laid by *D. suzukii* were excluded from the statistical analysis.

### 2.5. Statistical Analyses

We considered changes in the proportion of eggs individual flies laid into the different host categories during one egg-laying period as an adequate indicator of the dynamics in *D. suzukii* host choice (see also [11]). Thus, we utilized generalized linear models (GLMs) with a *binomial* distribution (whenever overdispersion was detected, *quasibinomial* was used instead) and a *logit* link function in version 4.0.2 of R, using R Studio [15]. We used backward elimination of statistically non-significant variables to obtain the most parsimonious model explaining the results of our experiments. We present these minimum adequate models and the respective analysis of variance (ANOVA) results. Statistically non-significant factors plus their *p*-values are mentioned. Additionally, we specified GLMs with a *poisson* distribution and *log* link function to test whether the individual overall reproductive output, i.e., total number of eggs laid, differed between treatments. All graphs were prepared in *ggplot2* using the R Studio environment.

The regular binomial GLM tests whether the intercept differs from 0.5. As we manipulated the proportions of treated/intact fruits in Experiment 1, we had different expectations regarding the distribution of eggs if the flies distributed them randomly across the twelve fruits at each individual proportion of intact fruits, namely 0.5, 0.33, and 0.17. We used the *offset* argument in R to fix a new intercept, namely *logit* (1/3) and *logit* (1/6), for the corresponding expected proportion of eggs in intact fruits. This allowed us to explicitly test whether we could reject the null hypothesis, i.e., the flies have no preference. The raw data of all experiments are accessible in the Appendix A.

## 3. Results

### 3.1. Experiment 1: Drosophila suzukii Egg-Laying Response to Fruits with Heterospecific Egg-Laying Cues

Whether *D. suzukii* were exposed to an environment that contained fruits with heterospecific egg-laying cues or wounded fruits only did not influence the total number of eggs laid (Poisson GLM, Type II ANOVA: χ^²^ = 2.148, d.f. = 3, *p* = 0.542); also, the variation in the relative abundance of intact blueberries had no effect on the total reproductive output during one egg-laying period (Poisson GLM, Type II ANOVA: χ^²^ = 0.252, d.f. = 1, *p* = 0.615). However, the way the flies distributed their eggs across the different fruit categories was significantly affected by variation in the relative abundance of intact fruits (Logistic GLM, Type II ANOVA: χ^²^ = 46.20, d.f. = 3, *p* < 0.001) and fruit treatment (Logistic GLM, Type II ANOVA: χ^²^ = 78.61, d.f. = 1, *p* < 0.001) (Figure 1). The total number of eggs laid by one female had no statistically significant effect on the distribution of eggs into intact fruits (Logistic GLM, Type II ANOVA: χ^²^ = 1.204, d.f. = 1, *p* = 0.273). The higher the proportion of intact fruits, the more eggs the flies laid in them (Figure 1). Importantly, for all combinations of treatment and proportion of intact fruits, we found a significant deviation from the expected proportion of eggs in intact fruits if the flies had not responded to the different fruit categories (*p* < 0.001 for all combinations; Appendix B: Table A1). The flies’ egg-laying response shows a clear preference for intact fruits. A post-hoc test without the results of the “wounding only” treatment revealed no species-specific response of *D. suzukii* to cues from heterospecific flies (Logistic GLM, Type II ANOVA: χ^²^ = 4.146, d.f. = 2, *p* = 0.126), while the significant effect of the variation in the relative abundance of intact fruits persisted (Logistic GLM, Type II ANOVA: χ^²^ = 41.975, d.f. = 1, *p* < 0.001). Therefore, the preference for intact fruits was stronger in treatments where intact fruits were combined with fruits containing heterospecific egg-laying cues than with wounded fruits only (Figure 1).

### 3.2. Experiment 2: Drosophila suzukii Egg-Laying Response to Heterospecific and Fruit-Wounding Cues

When offered together with wounded fruits, *D. suzukii* females laid ~70% of their eggs into intact fruits (Figure 2A), which was significantly different from the expected 1:1 distribution if the flies had no preference for either fruit category. In combination with intact fruits but previously exposed to *D. melanogaster*, ~72% of the eggs were laid into intact and non-exposed fruits (Figure 2B). However, intact but exposed blueberries were preferred (~69%) over wounded ones (Figure 2C). Lastly, *D. suzukii* preferred wounded only over wounded and exposed berries (Figure 2D).

## 4. Discussion

*Drosophila suzukii* showed a clear oviposition preference for intact and uninfested blueberries over berries that contained egg-laying cues of different *Drosophila* species. Cues associated with the heterospecific infestation of wounded fruits appear to strongly determine the hierarchical order of preferences between different host fruit categories: even when the relative abundance of cue-containing and wounded fruits was more than 80%, *D. suzukii* still preferred the few fresh and intact fruits (Experiment 1). As we removed the heterospecific eggs, avoidance of oviposition can be induced by cues that act independently of the actual presence of the eggs. This assumption was supported by our observations that intact fruits previously exposed to *D. melanogaster* visits without egg laying were also avoided by *D. suzukii* (Experiment 2). Chemical cues associated with feces deposited on the fruit skin by heterospecific flies, possibly containing insect pheromones [16] and waste products [17] or metabolites of gut microbes [18], may ultimately reduce the tendency of *D. suzukii* to oviposit in or even visit *D. melanogaster*-exposed blueberries.

Our results verify the previous observation [11] that *D. suzukii* tend to avoid oviposition into wounded blueberries. In our experiment, wounding and heterospecific cues seemed to additively or synergistically interact to influence the oviposition, as the combination of these cues significantly reinforced *D. suzukii* egg-laying avoidance (Experiment 2). Interestingly, intact and *D. melanogaster*-exposed fruits were preferred when offered together with wounded but non-exposed fruits. This suggests that, in the process of host fruit acceptance/avoidance, cues that indicate wounded fruits rank higher in the cue hierarchy of *D. suzukii* oviposition decisions than those that indicate the presence of hetero-specifics (Figure 3). The nature of the wounding cue is unknown. Thus, we cannot elucidate whether flies respond to this cue during the searching phase prior to contact with a fruit or the post-alighting contact phase. Tactile and gustatory stimuli due to texture and consistency differences [19,20] between wounded and intact fruits may be perceived during the contact phase. Chemical, e.g., wound-activated, emission of plant compounds [21] might also play a role, which could even be perceived by flies from a distance. Although the nature of the cues perceived by the flies still need to be identified, our data show how important the “wounding factor” is in individual *D. suzukii* egg-laying decisions. The observed responses may even expand to behaviors beyond the fruit-patch level and affect the flies’ searching patterns and hence the dispersal of females within and between fruiting bushes/trees.

Thus far, the properties associated with fruit injury have not been identified as a critical cue for *D. suzukii* host selection. This has mainly been due to the fact that most studies have used an artificial substrate to quantify oviposition decisions. Despite the need to reduce the natural environmental complexity to identify the key properties that drive *D. suzukii* host fruit selection in the field, “overreduction” may lead to wrong or incomplete conclusions, which could hamper the development of sustainable management tools. For example, given that egg-laying *D. suzukii* females rank wounded fruits generally lower than those associated with intact fruits (Figure 3), identification of the wounding cues may help to develop repellents that improve push–pull strategies [22]. Our experimental approach also includes manipulation of the relative abundance of certain host fruit categories, which allows us to test for the sensitivity of egg-laying responses along environmental gradients that induce variation in time limitation [11]. Knowing such reaction norms (sensu lato) in egg laying can inform us about how reliable the application of particular cues in repelling or deterring *D. suzukii* will be in different contexts [22], e.g., seasonal variation in the availability of different host categories. In this regard, we need to explore further whether the proposed host selection hierarchy (Figure 3) holds under, for example, changes in fly age, population densities, and fruit varieties.

Previous studies using different fruits, e.g., peaches or grapes, found an apparent preference for wounds [23,24]. However, the skins of those fruits, as determined by their firmness or hairiness, seem to be a significant barrier to successful oviposition [25]; either they are impenetrable or the time investment to eventually create an oviposition site is too high, and in consequence flies deposit their eggs in fruits wounds instead. This does not seem to be the case for oviposition into blueberries. Therefore, our study highlights the strong preference of *D. suzukii* for intact and healthy fruits; however, this preference may be masked if flies only have access to damaged fruits with impenetrable or otherwise deterring skin.

## 5. Conclusions

Our individual-based, semi-natural approach provides evidence that *D. suzukii* females use both fruit wounding and heterospecific cues to avoid egg laying into already infested fruits. It is the behavioral responses to these inevitably interacting cues that reveal a host fruit preference hierarchy in *D. suzukii*. The avoidance of egg laying into fruits containing cues that indicate the presence of heterospecifics may prove beneficial if *D. suzukii* larvae are poor competitors [5]. We hypothesize that such aversive behavior is critical for the host fruit preference hierarchy, which may influence individual dispersal and hence the expansion of *D. suzukii* populations.

## Figures and Tables

**Figure 1 insects-12-00424-f001:**
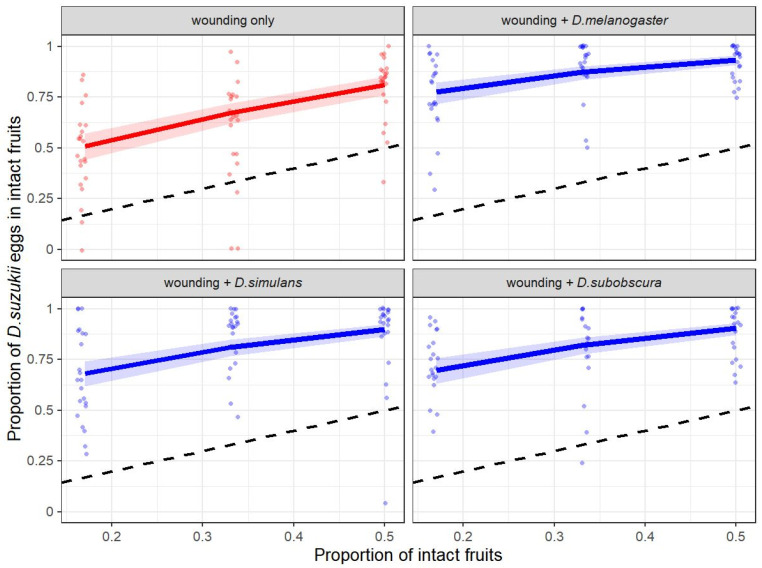
Proportion of eggs laid by *Drosophila suzukii* into fresh and intact blueberries, offered together with fruits that were wounded only or wounded and contained egg-laying cues of different *Drosophila* species (*D. melanogaster*, *D. simulans*, *D. subobscura*); prior to the use of the blueberries in the choice experiments the heterospecific eggs were removed. Intact and fresh fruits were embedded in a matrix of wounded/infested blueberries (a total of 12 berries) that varied in their relative abundance (*x*-axis), i.e., 2 intact vs. 10 wounded/infested (0.17), 4/8 (0.33), 6/6 (0.5). For better visibility, the data points are staggered around the corresponding values on the *x*-axis. The solid and colored lines represent the predicted values from the Logistic GLM. The shaded areas show the standard error of the predicted model. The dashed lines represent the expected proportion of *D. suzukii* eggs in fresh and intact fruits if the female flies had no preference for either host fruit category and distributed their eggs randomly across the twelve blueberries.

**Figure 2 insects-12-00424-f002:**
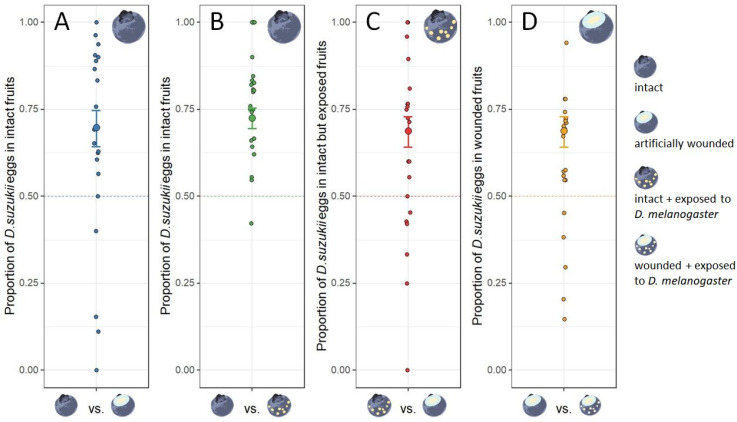
Proportion of eggs laid by *Drosophila suzukii* into specific target fruit category (*y*-axis) when combined with a different fruit category (*x*-axis): (**A**) target category: intact; different category: wounded; (**B**) target category: intact; different category: exposed; (**C**) target category: exposed; different category: wounded; (**D**) target category: wounded; different category: wounded and exposed. For each replicate, the relative abundance was 0.5 in a matrix of 12 blueberries. The dashed lines represent the expected proportion of *D. suzukii* eggs in the target fruit category if the female flies had no preference for either category and distributed their eggs randomly across the twelve blueberries. The predicted values ± standard errors from the corresponding GLMs are shown. In all combinations, the flies laid a significantly higher proportion of eggs in the target fruit category than expected under the null hypothesis, i.e., flies do not have any preference: (**A**) *t* = 3.37, *p* = 0.003; (**B**) *t* = 6.53, *p* < 0.001; (**C**) *t* = 3.87, *p* < 0.001; (**D**) *t* = 2.33, *p* = 0.030.

**Figure 3 insects-12-00424-f003:**
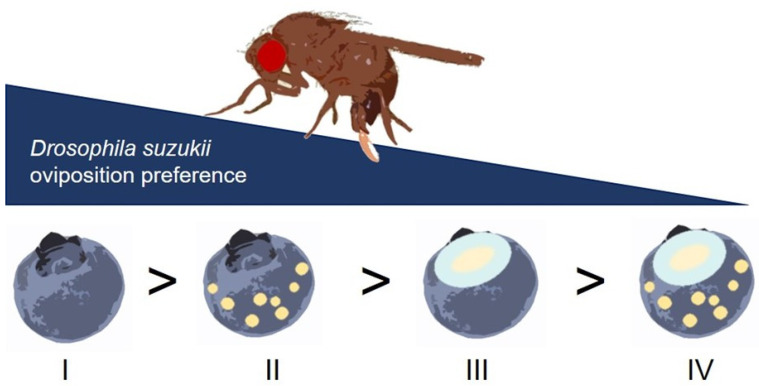
Host fruit selection hierarchy in *Drosophila suzukii* on blueberries. The proportion of eggs laid into different potential host fruit categories revealed an oviposition preference for intact fruits (**I**) over intact ones carrying cues, probably fecal material, of heterospecific flies (**II**), followed by wounded fruits (**III**) and those that were wounded and had heterospecific marks (**IV**).

## Data Availability

Data generated during the experiments described in the method section are available as Appendix A Appendix A: “Raw Data.xlsx”.

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
