# Peer review of "Mind the Wound!—Fruit Injury Ranks Higher than, and Interacts with, Heterospecific Cues for Drosophila suzukii Oviposition"

_insects, 2021, doi:10.3390/insects12050424_

Round 1

Reviewer 1 Report

This manuscript describes the oviposition preference of Drosophila suzukii for different berries quality (intact, wounded, or contaminated by heterospecific). The subject is well explained in the introduction and the literature is adequately cited. The data presented here are interesting and clarify one aspect of D. suzukii behaviour but is limited to explain the ecology of the pest. This research appears to me with a limited potential of citation.

However, whatever of the editorial decision, I have very minor comments to improve the manuscript:

Line 124: Are females mated or virgins for different species? Because if females are not mated, this could probably change their behavior. How did you control this state? How many times the experiment was running? Please add information as well as for experiment 2.

Line 130-In experiment 1 this is not clear for me, I understand you infested wounded blueberry by heterospecific eggs but I understand you removed the eggs. If yes, then in line 140 replace “infested” with “cues from” or another term. Because sentence sounds like if eggs are still in the berries. Same in figure 1, I would change the term “infested” to be sure that the reader does not have in mind that eggs are still present in the berries.

Could you explain why you tested three competitors in experiment 1 and subsequently only one in experiment 2. How did you choose the species and why D. subobscura/D. simulans were not tested in experiment 2. Do you suspect D. suzukii different response depending on the identity of the heterospecific species?

Figure 3: I would not recommend adding category IV because you have not tested all 4 categories together. Only intact compared to wounded + heterospecific mark in experiment 1 and experiment 2 only 3 categories were tested. For example, you have not tested category II over category IV. Please remove.

Discussion: Could you discuss your results in light of findings on wounded grapes preference by D. suzukii (for example, but there is other manuscripts: https://www.mdpi.com/2075-4450/10/12/432). In the case of grapes, wounded fruits are prefered, so repellency of wounded fruits could not be generalized for D. suzukii I think. The title of the manuscript should be changed.

Considering the small length of the discussion, I’m not convinced of the necessity of a conclusion.

Author Response

This manuscript describes the oviposition preference of Drosophila suzukii for different berries quality (intact, wounded, or contaminated by heterospecific). The subject is well explained in the introduction and the literature is adequately cited. The data presented here are interesting and clarify one aspect of D. suzukii behaviour but is limited to explain the ecology of the pest. This research appears to me with a limited potential of citation.

Authors' response: thanks for the positive evaluation of our paper

However, whatever of the editorial decision, I have very minor comments to improve the manuscript:

Line 124: Are females mated or virgins for different species? Because if females are not mated, this could probably change their behavior. How did you control this state? How many times the experiment was running? Please add information as well as for experiment 2. 

Authors' response: as described in the method section, for each species, we had a population of approx. 200 flies; we added information of fly age. It was a mixed sex population, and as the flies laid eggs, females must have been mated. We think it is enough to mentioned that flies laid eggs to clarify the experimental design in this regard. The number of replication is already given in the method section.

Line 130-In experiment 1 this is not clear for me, I understand you infested wounded blueberry by heterospecific eggs but I understand you removed the eggs. If yes, then in line 140 replace “infested” with “cues from” or another term. Because sentence sounds like if eggs are still in the berries. Same in figure 1, I would change the term “infested” to be sure that the reader does not have in mind that eggs are still present in the berries.

Authors' response: we agree with the reviewer's view that this may cause confusion and changed the text accordingly (see track changes version); basically we changed "infested fruits" to "fruits containing egg-laying cues" wherever possible.

Could you explain why you tested three competitors in experiment 1 and subsequently only one in experiment 2. How did you choose the species and why D. subobscura/D. simulans were not tested in experiment 2. Do you suspect D. suzukii different response depending on the identity of the heterospecific species?

Authors' response: some information on that is already given in the method section; we added some more explanatory sentences 

Figure 3: I would not recommend adding category IV because you have not tested all 4 categories together. Only intact compared to wounded + heterospecific mark in experiment 1 and experiment 2 only 3 categories were tested. For example, you have not tested category II over category IV. Please remove.

Authors' response: here, the reviewer raised an interesting point. In response to the reviewer's concern, we ran another experiment that quantified the D. suzukii's egg distribution when given the choice between wounded and wounded plus heterospecific cues. The flies showed a preference for the wounded only fruits. Thus, our argument holds true, and we therefore keep category IV in the proposed egg-laying hierarchy. We added a new figure 2D, and updated raw data file accordingly.

Discussion: Could you discuss your results in light of findings on wounded grapes preference by D. suzukii (for example, but there is other manuscripts: https://www.mdpi.com/2075-4450/10/12/432). In the case of grapes, wounded fruits are prefered, so repellency of wounded fruits could not be generalized for D. suzukii I think. The title of the manuscript should be changed.

Authors' response: that's an interesting aspect. To be able to infer preference from a choice assay requires that both fruit categories have the same accessibility for the flies to deposit their eggs. Grapes are known for not being super susceptible to D. suzukii as a rather tough fruit skin prevents oviposition, and then of course the flies seem to prefer the damaged ones; but this an only apparent preference, as flies are simply excluded to a large extent from fruits with very firm skin. We added a paragraph discussing that issue, but won't change the title, according to the misleading conclusion of an only apparent preference.

Considering the small length of the discussion, I’m not convinced of the necessity of a conclusion.

Authors' response: well, we are convinced!

Reviewer 2 Report

This study explored potential effect of “expected interspecific competition” on the host selection or acceptance by SWD female. The results showed that female SWD prefers to lay eggs into intact over wounded or previously exposed fruit to other drosophila species. The paper is well written; the experiments were carefully designed and well performed, and the data are clearly presented. I do not see any issues.

My only comment is regarding this statement (L226-227): “Thus far, properties associated with fruit injury have not been identified as a critical cue for D. suzukii host selection”. I noticed in another study (Stewart et al. 20214), the authors showed that peach surface damage can facilitate D. suzukii oviposition, and the fly also tends to lay eggs in soft spots on the fruit, this is because intact peach fruit is hard to be penetrated. So, I guess this may depend on fruit species and maturation stage (or firmness). I am surprised that the fly does not prefer to lay eggs in wounded blueberry fruit as this will save oviposition time although it may face interspecific competition.

Stewart et al. (2014) Factors limiting peach as a potential host for spotted wing drosophila. Journal of Economic Entomology 107:1771-1779.

L122: the flies did not lay eggs (flies) but defecated on these fruits (delete flies)

Author Response

Authors' response: many thanks for the very positive response and evaluation of our paper

My only comment is regarding this statement (L226-227): “Thus far, properties associated with fruit injury have not been identified as a critical cue for D. suzukii host selection”. I noticed in another study (Stewart et al. 20214), the authors showed that peach surface damage can facilitate D. suzukii oviposition, and the fly also tends to lay eggs in soft spots on the fruit, this is because intact peach fruit is hard to be penetrated. So, I guess this may depend on fruit species and maturation stage (or firmness). I am surprised that the fly does not prefer to lay eggs in wounded blueberry fruit as this will save oviposition time although it may face interspecific competition.

Stewart et al. (2014) Factors limiting peach as a potential host for spotted wing drosophila. Journal of Economic Entomology 107:1771-1779.

Authors' response: good and interesting point, which now address in a new paragraph in the discussion section; the suggested reference is a good one, which we use in that part of the discussion.

L122: the flies did not lay eggs (flies) but defecated on these fruits (delete flies)

Authors' response: deleted